# Productivity of Services in the Countries of Central and Eastern Europe: Analysis Using Malmquist Indices

**Alejandro Alcalá-Ordóñez, Francisco Alcalá-Olid ***  **and Pablo Juan Cárdenas-García**

Department of Economics, University of Jaen, 23071 Jaén, Spain; aao00006@red.ujaen.es (A.A.-O.);
pcgarcia@ujaen.es (P.J.C.-G.)
* Correspondence: falcala@ujaen.es

**Abstract:** This research aims to study the growth of productivity in the service sector in the former Central and Eastern European Countries (CEECs) and their determinants. For this purpose, non-parametric frontier techniques were used to measure the variations in productivity and determine the explanatory factors of these changes in total factor productivity; the methodology of the Malmquist index with output orientation and its decomposition in technical change, pure technical efficiency and scale efficiency was used for the period 2000–2019. The results obtained indicate that the productivity of services in the most recently incorporated countries grew by 1.3 per 100 on average per year compared to 1.6 per 100 in manufacturing. The most important driver of such growth was found to be improvement in technical change (frontier shift) rather than improvement in efficiency.

**Keywords:** productivity; services; Malmquist index; DEA; non-parametric methods





## 1. Introduction

Much of the economic literature on the productivity of the service sector has emphasised its low level and even its slow increase. To the extent that in the long term the economic growth of a country depends on its productivity and its tertiary activities contribute more to the generation of the aggregate product, the ability to improve the welfare of the population is conditioned by what happens in this sector.

If long-term growth is based on the ability to increase productivity (Krugman 1990), given that services are gaining prominence and their productivity (or growth rate) is lower than that of the non-tertiary branches, the improvement in the well-being of the population will be less than what would be achieved if the yield of production per factor used were higher.

In recent years, the low productivity growth of the services sector has attracted the attention of numerous researchers, such that, as Cuadrado (2016) pointed out, the analysis of the sector from the macroeconomic point of view has increased. The traditional unproductive character that has been given to tertiary activities that was already advocated by classical economists such as Adam Smith or Jean Baptiste Say sometimes persists, as can be seen in the theses maintained not so long ago by Baumol (1967, 1986); Baumol et al. (1989) or Nusbaumer (1987), among others[1].

In fact, the general impression is that in all economies, the performance of the labour factor in the services sector advances slowly. As Maroto (2009) has rightly pointed out, this statement is not new, but rather it was already raised by Clark (1940) and Fourastié (1949) and, notably, in the works of Baumol, in which the most important progress was made on the relationship between the growth of services in the economy and their low productivity, coming to speak of "cost disease" in all tertiary activities.

In any case, years after their first work on this aspect, Baumol et al. (1989) qualified this thesis by distinguishing between different classes of services so that branches with slow growth coexist with others whose rate of increase in productivity even exceeds that of

industrial activities, reaching the conclusion that it could be argued that the aforementioned disease was considered "cured" (Cuadrado and Maroto 2007), since it maintains that only a third of the sector can be classified as having low productivity and/or slow growth.

The main criticisms that have been made about the aforementioned disease, following Maroto (2009) and Cuadrado and Maroto (2012), are based, in particular, on the role of innovation and knowledge in some services (Baumol 2000, 2001; Djellal and Gallouj 2008, 2010). Secondly, there are concerns about the indirect effects of services on other non-tertiary activities, as well as the problems derived from the difficulties in measuring this magnitude (Rubalcaba 1999; Wolff 1999; Kox 2002), raised by authors such as Gadrey (1996) or the European Commission (2004). Thirdly, there is the fact that productivity growth goes beyond the labour factor and is influenced by other factors (Cuadrado and Del Río 1993; Kox 2002), such as the nature of the service, the organization and segmentation of the market to which it is directed, or the possibilities of substitution between capital and labour. Fourthly, several authors have pointed out that Baumol's theories are only applicable to final services and not to those used as intermediate input. Furthermore, the low productivity of some services should be complemented with that obtained by the activities that use them as intermediate consumption (Fixler and Siegel 1999; Raa and Wolff 2001). The last factor indicated by Maroto (2009) is related to the high productivity shown by branches related to ICT[2], which would explain the increasing returns to scale (Wölfl 2003). Finally, Cuadrado and Maroto (2012) point out that the analysis from the macroeconomic point of view does not seem to be the most appropriate, with microeconomic ones being preferable (Lichtenberg 1995; Brynjolfsson and Hitt 1993; Pilat 2005; David 1990).

In this context, the main objectives pursued with this work are: to know what has been the change in the total productivity of the factors in the services sector of the economies of the countries of Central and Eastern Europe; analyse whether the magnitude of said change is explained by technological change or by the change in the efficiency with which the service is provided; and, finally, determine if the change in efficiency is caused by a change in scale efficiency or by the alteration in pure technical efficiency.

With this purpose, the rest of the work has the following structure. The Section 2 reviews the methods used to measure productivity and some of the contributions made related to services. Section 3 presents the data and the methodology. Section 4 is devoted to the results obtained and their discussion. Finally, our work closes with a section with the main conclusions.

## 2. Literature Review

Productivity is one of the most important factors of economic growth (Afsharian and Ahn 2015; Bongers and Torres 2020) and, therefore, its measurement has become the subject of numerous research works in recent years. Two types of basic models are usually used to measure productivity, on the one hand using econometric methods (for example, growth regressions, stochastic frontier analysis, SFA) and on the other, using non-parametric or deterministic methods, such as data envelopment analysis (DEA), the free disposal Hull model, growth accounting or efficiency indices (Carlaw and Lipsey 2003; Del Gatto et al. 2011).

In this work, we chose a non-parametric model to study the evolution of the productivity of the service sector in the countries of Central and Eastern Europe. There are numerous techniques that can be used to measure change in productivity levels. Productivity measures can be classified as single factor (single output and single input), as well as measures of multifactorial productivity (when there is one output and several inputs)[3]. The inputs used in the different jobs to measure productivity are usually the labour factor, capital, energy, raw materials, etc. (Zrelli et al. 2020).

Caves et al. (1982) introduced the Malmquist index to measure changes in the relative productivity of different decision making units (DMUs) at different moments in time[4]. These DMUs can be companies, manufacturing plants, utilities, countries or regions. There are two fundamental reasons that justify the importance given to this index. The first is

that the Malmquist index (MI) is based on the formulation of technical efficiency, therefore it only requires information related to the inputs and outputs used. The second is that said index can be broken down into several components, which facilitates the interpretation of the changes experienced by each DMU[5].

Since the publication of the work by Grifell-Tatjé and Lovell (1995), which demonstrated the effectiveness of this technique to measure the change in productivity levels, the use of the Malmquist index to measure said changes in different areas has gradually increased.

Various studies have used the Malmquist index to assess the change in total factor productivity. Without intending to be exhaustive, reference will be made to some investigations that have used the MI in different economic activities, territorial and temporal areas in order to show the versatility of this tool and its full adaptability to be used in the context of this study, that is, in the set of activities in the services sector of an economy.

In recent years, we can find works such as that of Madden and Savage (1999) on the telecommunications sector in 74 countries during the period 1991–1995. Mahlberg and Url (2003) applied MI to the Austrian insurance sector. Odeck (2008b) used it in the study of the effects on productivity of the merger processes of public transport companies in Norway, while in Odeck (2008a) what is analysed is the change experienced by toll companies in said country. Yu (2008) dedicated his research to the evolution of productivity in the bus transportation system in Taiwan. Yu and Hsu (2012), also in Taiwan, focused their work on air transport. Frančeškin and Bojnec (2022) used DEA and the Malmquist index for the analysis of the productivity of the hotel sector in Slovenia. Cullmann and Von Hirschhausen (2008) focused their attention on electricity supply companies in Poland. The banking sector has also been the object of analysis through this methodology, as is the case, for example, of Grifell-Tatjé and Lovell (1996), Portela and Thanassoulis (2010) or Shair et al. (2021). In the field of the health sector, we find numerous works, such as those of Färe et al. (1997) or Prior (2006). In the educational sector, an appreciable number of contributions have also been made, such as those of Thanassoulis et al. (2011), Parteka and Wolszczak-Derlacz (2013), and Rong et al. (2018). In short, all activities included in the services sector have been analysed under the non-parametric methodology of the DEA and the Malmquist index in order to study how productivity has behaved and what have been the factors that have affected it or led to its upward or downward modification.

Despite the numerous works that exist, both at the country level and the regional level, for the different branches that make up the service sector, the work presented here aims to cover a plot little treated by the economic literature in all these years. Specifically, there are very few works (or none have been found) that analyse the evolution of the productivity of the service sector as a whole, as was performed in this work. In this sense, it should be noted that the main references in this case are those corresponding to the works of Maroto (2009) and Cuadrado and Maroto (2012). The first of them, in which the change in productivity of the services sector in a group of EU countries was analysed, together with the United States, served as a reference to carry out the study presented here on said productivity in the bloc of countries that joined the EU during this century.

## 3. Data and Methodology

### 3.1. Data Description

The analysis of productivity in the services sector, its evolution and the factors that explain its behaviour in the analysis period for the group of Central and Eastern European countries that joined the EU in this century was carried out based on the data published by Eurostat, which were complemented with the EU KLEMS database. From there, quantitative information was obtained on the values of production, employment, capital and, consequently, productivity for all the territorial areas on which we focused this work. The aforementioned variables can be broken down into 10 economic activities, according to the A10 breakdown contained in Commission Regulation 715/2010 of 10 August 2010. The activities mentioned are: agriculture, forestry and fishing (section A, according to equiv-

alent NACE Rev. 2 coding); mining and quarrying, industrial manufacturing, electricity, gas, steam and air conditioning supply, water supply, sewerage, waste management and remediation activities (sections B, C, D and E); construction (F); wholesale and retail trade, repair of motor vehicles and motorcycles, transportation and storage, accommodation and food service activities (G, H and I); information and communications (J); financial and insurance activities (K); real estate activities (L); professional, scientific and technical activities, administrative and support service activities (M and N); public administration, defence, education, human health and social work activities (O, P and Q); and arts, entertainment and recreation, other service activities, and activities of household and extra-territorial organizations and bodies (R, S, T and U). This subdivision allowed us, as will be seen at the end of the results section, to add the values corresponding to the industrial and service sectors, excluding those corresponding to the primary, mining, energy and water, and construction sectors. In this way, it is possible to analyse the evolution of the productivity of service activities in comparison with the industrial sectors (manufacturing, in the strict sense) and the economy as a whole.

The period of analysis covered the years 2000–2019, both inclusive. The limited availability of data for the years 2020 and 2021 in an appreciable number of countries, as well as the distortion that both years would introduce into the research as a consequence of the pandemic, were the main causes for closing the study period in the aforementioned 2019. However, the possibility of expanding this work is already being assessed, when the data is fully available, in order to see the impact that COVID-19 may have had on the evolution of the productivity of the services as a whole of the EU. In this investigation, it was decided, with respect to the work factor, to use the number of hours worked instead of the number of workers, since as Cuadrado and Maroto (2012) point out, this takes into account changes in the full–part-time ratio of workers[6].

*3.2. Methodological Issues*

The initial DEA model, based on the earlier work of Farrell (1957), was proposed by Charnes et al. (1978) and it is known as the CCR model. It is a non-parametric method in which multiple inputs and outputs can be assessed to measure the relative efficiency of a group of companies and it allows us to model production technology without imposing a particular functional form. The DEA-CCR model provides a measurement of technical efficiency[7], assuming that the DMU operates under constant returns to scale. The formalization of the mathematical optimization programme, according to the basic model (output orientated[8]) proposed by Charnes et al. (1978), is as follows

$$\min q = \sum_{i=1}^{m} v_i x_{i0} \tag{1}$$

subject to

$$\sum_{i=1}^{m} v_i x_{ij} - \sum_{r=1}^{s} \mu_r y_{rj} \leq 0$$
$$i = 1, 2 \ldots m;$$
$$\sum_{r=1}^{s} \mu_r y_{r0} = 1 \qquad r = 1, 2 \ldots s;$$
$$\mu_r, v_i \geq \varepsilon > 0 \qquad j = 1, 2 \ldots n;$$

In this programme, we are dealing with a vector of N companies that produce s outputs y using m inputs x. The variables $\mu_r$ and $v_i$ represent the "weights" corresponding to outputs and inputs, respectively, and $\varepsilon$ is a non-Archimedean infinitesimal that, as Ali and Seiford (1993) point out, cannot be represented as a real number, while q represents the efficiency value of the io–th unit.

This can be transformed into envelopment form as follows

$$\max \phi + \varepsilon \left( \sum_{i=1}^{m} s_i^- + \sum_{r=1}^{s} s_r^+ \right) \tag{2}$$

subject to

$$
\begin{aligned}
\sum_{j=1}^{n} x_{ij}\lambda_j + s_i^- &= x_{i0} & i &= 1, 2 \ldots m; \\
\sum_{j=1}^{n} y_{rj}\lambda_j - s_r^+ &= \phi y_{r0} & r &= 1, 2 \ldots s; \\
\lambda_j &\geq 0 & j &= 1, 2 \ldots n;
\end{aligned}
$$

where $x_{ij}$ and $y_{rj}$ are the inputs and outputs, respectively, and $si^-$ and $sr^+$ are the corresponding slacks. In order for a decision-making unit to be considered efficient, it has to meet the following conditions (Cooper et al. 2002): (a) the efficiency score has to be equal to one ($\phi^* = 1$), and (b) all slacks have to be equal to zero. If the constraint $\sum_{j=1}^{n} \lambda_j = 1$, is added to the model, then it is called a BCC model (Banker et al. 1984), which assumes variable returns to scale. This added constraint introduces an additional variable $\mu 0$ into the dual problem, which allows us to identify the nature of returns to scale (increasing, constant or decreasing).

These models are frequently used for measuring relative efficiency of a set of firms under static conditions. To evaluate the efficiency change of a firm between two time periods, we calculated the Malmquist productivity index.

The Malmquist productivity index, introduced by Caves et al. (1982), evaluates the productivity change of a decision-making unit between two time periods by calculating the ratio of the distances of each time period relative to a common technology. If the reference technology is period $t + 1$, the Malmquist (output orientated) index between period $t$ and period $t + 1$ can be defined as follows

$$
M_O^{t+1} = \frac{D_O^{t+1}(x_{t+1}, y_{t+1})}{D_O^{t+1}(x_t, y_t)} \tag{3}
$$

However, if the reference technology is period $t$, it can be written

$$
M_O^{t} = \frac{D_O^{t}(x_{t+1}, y_{t+1})}{D_O^{t}(x_t, y_t)} \tag{4}
$$

In the above equations, M is the value of Malmquist index, the subscript O indicates that the framework is output-orientated, $D(x, y)$ is the distance function and x and y are the input and output vectors, respectively. To avoid the arbitrariness of choosing one of the two technologies, Färe et al. (1989, 1992) defined the index as the geometric means of Equations (3) and (4), i.e.,

$$
M_O(y_{t+1}, x_{t+1}, y_t, x_t) = \left[ \frac{D_O^t(x_{t+1}, y_{t+1})}{D_O^t(x_t, y_t)} \frac{D_O^{t+1}(x_{t+1}, y_{t+1})}{D_O^{t+1}(x_t, y_t)} \right]^{1/2} \tag{5}
$$

From this, the index was analysed and decomposed in several ways and parametric and non-parametric techniques have been used to calculate it[9]. Equation (5) can be rearranged as

$$
M_O(y_{t+1}, x_{t+1}, y_t, x_t) = \frac{D_O^{t+1}(x_{t+1}, y_{t+1})}{D_O^t(x_t, y_t)} \left[ \frac{D_O^t(x_{t+1}, y_{t+1})}{D_O^{t+1}(x_{t+1}, y_{t+1})} \frac{D_O^t(x_t, y_t)}{D_O^{t+1}(x_t, y_t)} \right]^{1/2} \tag{6}
$$

In Equation (6), we observe that the ratio outside the square brackets measures the change in the output-orientated technical efficiency between periods $t$ and $t + 1$. This is called the catch-up effect, and indicates the degree to which a DMU improves (if its value is greater than one) or worsens (less than one) its efficiency. The remaining part of Equation (6) measures technical change (frontier-shift effect) and reflects the change in the efficient frontiers between the two time periods. The first ratio inside the square brackets measures the frontier-shift effect in period $t + 1$ and the second one in period $t$. If the

frontier-shift effect is greater than one (equal or less than one), this indicates progress (the status quo or regress) in the frontier technology around DMU from period t to t + 1. The Equation (6), in abbreviated form, can be expressed as:

$$M_O = TEC \cdot TC \tag{7}$$

where TEC means technical efficiency change and TC means technical change.

According to Färe et al. (1994), the ratio that measures the change in technical efficiency (the first ratio of the Equation (6)) can be decomposed into a pure technical efficiency change component (measured relative to the arguably true variable returns to scale frontier) and a scale efficiency change component, as follows:

$$PETC = \frac{D_{OV}^{t+1}(x_{t+1}, y_{t+1})}{D_{OV}^{t}(x_t, y_t)} \tag{8}$$

$$SEC = \frac{\frac{D_{OC}^{t+1}(x_{t+1}, y_{t+1})}{D_{OV}^{t+1}(x_{t+1}, y_{t+1})}}{\frac{D_{OC}^{t}(x_t, y_t)}{D_{OV}^{t}(x_t, y_t)}} \tag{9}$$

where PETC means pure technical efficiency change, SEC means scale efficiency change, the subscript V relates to variable returns to scale and subscript C relates to constant return to scale. Therefore, the output-orientated Malmquist index can be written as

$$M_O = PETC \cdot SEC \cdot TC \tag{10}$$

or in a more formal expression

$$M_O(y_{t+1}, x_{t+1}, y_t, x_t) = \frac{D_{OV}^{t+1}(x_{t+1}, y_{t+1})}{D_{OV}^{t}(x_t, y_t)} \frac{\frac{D_{OC}^{t+1}(x_{t+1}, y_{t+1})}{D_{OV}^{t+1}(x_{t+1}, y_{t+1})}}{\frac{D_{OC}^{t}(x_t, y_t)}{D_{OV}^{t}(x_t, y_t)}} \left[ \frac{D_{OC}^{t}(x_{t+1}, y_{t+1})}{D_{OC}^{t+1}(x_{t+1}, y_{t+1})} \frac{D_{OC}^{t}(x_t, y_t)}{D_{OC}^{t+1}(x_t, y_t)} \right]^{1/2} \tag{11}$$

To calculate Equation (11), we must solve the six component distance functions, which involve the next six linear programming problems:

$$\left[ D_{OC}^{t}(x_t, y_t) \right]^{-1} = \max_{\phi, \lambda} \phi,$$

subject to

$$
\begin{aligned}
-\phi y_{it} + Y_t \lambda &\geq 0, \\
x_{it} - X_t \lambda &\geq 0, \\
\lambda &\geq 0
\end{aligned}
\tag{12}
$$

$$\left[ D_{OC}^{t+1}(x_{t+1}, y_{t+1}) \right]^{-1} = \max_{\phi, \lambda} \phi,$$

subject to

$$
\begin{aligned}
-\phi y_{i,t+1} + Y_{t+1} \lambda &\geq 0, \\
x_{i,t+1} - X_{t+1} \lambda &\geq 0, \\
\lambda &\geq 0
\end{aligned}
\tag{13}
$$

$$\left[ D_{OC}^{t}(x_{t+1}, y_{t+1}) \right]^{-1} = \max_{\phi, \lambda} \phi,$$

subject to

$$
\begin{aligned}
-\phi y_{i,t+1} + Y_t \lambda &\geq 0, \\
x_{i,t+1} - X_t \lambda &\geq 0, \\
\lambda &\geq 0
\end{aligned}
\tag{14}
$$

$$\left[ D_{OC}^{t+1}(x_t, y_t) \right]^{-1} = \max_{\phi, \lambda} \phi,$$

subject to

$$\begin{aligned}
-\phi y_{it} + Y_{t+1}\lambda &\geq 0, \\
x_{it} - X_{t+1}\lambda &\geq 0, \\
\lambda &\geq 0 \\
\left[ D_{OC}^{t}(x_t, y_t) \right]^{-1} &= \max_{\phi,\lambda} \phi,
\end{aligned} \tag{15}$$

subject to

$$\begin{aligned}
-\phi y_{it} + Y_{t}\lambda &\geq 0, \\
x_{it} - X_{t}\lambda &\geq 0, \\
\overrightarrow{1}\lambda = 1\lambda &\geq 0 \\
\left[ D_{OC}^{t+1}(x_{t+1}, y_{t+1}) \right]^{-1} &= \max_{\phi,\lambda} \phi,
\end{aligned} \tag{16}$$

subject to

$$\begin{aligned}
-\phi y_{i,t+1} + Y_{t+1}\lambda &\geq 0, \\
x_{i,t+1} - X_{t+1}\lambda &\geq 0, \\
\overrightarrow{1}\lambda &= 1 \\
\lambda &\geq 0
\end{aligned} \tag{17}$$

## 4. Results and Discussion

In this section, as previously indicated, the growth of productivity and its components, that is, technical efficiency and technical change, are estimated for the years from 2000 to 2019, both inclusive.

The first thing to be presented is the average growth rates of the three variables that were considered in the model for the set of the so-called former Central and Eastern European countries (CEECs), whose incorporation into the EU occurred in the present century. In Table 1, it can be seen how the growth of production during the indicated period of time was found to be 4.71 per 100 on average[10]. Above average, Malta (6.35 per 100) and Lithuania (6.02) stand out, followed by Romania (5.80), Slovakia (5.65), Poland (5.37), Estonia (4.97) and Latvia (4.88 per 100), the rest being below the average for this group of countries. In addition, the rest of the members of the EU only achieved an average growth of 1.44 per 100, which rises to 1.68 per 100 when taking into account all the member states belonging to the current European Union. An additional issue is the fact that in Romania, Latvia and Lithuania, the labour factor was reduced in this period.

**Table 1.** Evolution of production growth, labour factor and capital stock, 2000–2019. (Average annual growth rate, in %).

| Country/Area | Production | Employment | Capital |
|:---:|:---:|:---:|:---:|
| Bulgaria | 4.70 | 0.50 | 0.90 |
| Croatia | 2.36 | 0.25 | 5.81 |
| Cyprus | 3.22 | 1.78 | 3.31 |
| Czech | 3.84 | 0.26 | 2.90 |
| Estonia | 4.97 | 0.04 | 6.76 |
| Hungary | 3.31 | 0.11 | 1.51 |
| Latvia | 4.88 | −0.79 | 0.54 |
| Lithuania | 6.02 | −0.02 | 3.58 |
| Malta | 6.35 | 3.12 | 2.32 |
| Poland | 5.37 | 0.44 | 4.39 |
| Romania | 5.80 | −1.15 | 2.58 |

**Table 1.** *Cont.*

| Country/Area | Production | Employment | Capital |
|:---:|:---:|:---:|:---:|
| Slovakia | 5.65 | 0.66 | 1.78 |
| Slovenia | 3.20 | 0.37 | 0.91 |
| CEECs | 4.71 | 0.01 | 2.99 |
| EU14 | 1.44 | 0.43 | 1.32 |
| EU27 | 1.68 | 0.32 | 1.46 |

Source: Eurostat and EU KLEMS. Own elaboration.

The evolution of production factors, as can be seen, is different in the two groups of countries. Thus, while employment grew faster in the group of states that were in the EU for the longest time (0.43 compared to 0.01 per 100), in the case of capital, it was exactly the opposite, yielding an average annual rate of 2.99 per 100 in the eastern group of countries compared to 1.32 in the western ones. In this regard, it is worth noting the strong increase in the number of hours worked in Malta and Cyprus, as well as the sharp acceleration experienced by the capital stock in Estonia, Croatia and Poland.

This work's aim of analysing the productivity over the period of years studied was carried out through the calculation of the Malmquist indices and the components that integrate it, such as the change in efficiency, the change in scale and the technical change, for each of the countries studied. Since these are discrete values, each country had an index for each pair of years, that is, an index was obtained between one year and the next, and so on. Given that 20 years and 15 countries yield a large number of data, we considered using the mean values in order to simplify the information analysed, as can be seen in Table 2. The calculations of the Malmquist indices and their components were carried out with R software (Coll-Serrano et al. 2022).

**Table 2.** Malmquist index and its components, 2000–2019. (average annual growth).

| Country/Area | MI | TEC | TC | PETC | SEC |
|:---:|:---:|:---:|:---:|:---:|:---:|
| Bulgaria | 1.025 | 1.041 | 0.984 | 1.033 | 1.008 |
| Croatia | 0.986 | 0.996 | 0.990 | 0.997 | 0.999 |
| Cyprus | 1.002 | 1.000 | 1.002 | 1.000 | 1.000 |
| Czech | 1.010 | 1.010 | 1.000 | 1.008 | 1.003 |
| Estonia | 1.002 | 1.003 | 1.000 | 0.994 | 1.009 |
| Hungary | 1.017 | 1.019 | 0.998 | 1.015 | 1.005 |
| Latvia | 1.034 | 1.037 | 0.997 | 1.037 | 1.000 |
| Lithuania | 1.020 | 1.023 | 0.997 | 1.023 | 1.000 |
| Malta | 1.022 | 1.019 | 1.003 | 1.000 | 1.019 |
| Poland | 1.009 | 1.017 | 0.993 | 1.005 | 1.012 |
| Romania | 1.018 | 1.035 | 0.984 | 1.020 | 1.015 |
| Slovakia | 1.025 | 1.024 | 1.001 | 1.026 | 0.997 |
| Slovenia | 1.019 | 1.016 | 1.003 | 1.015 | 1.001 |
| CEECs | 1.015 | 1.018 | 0.996 | 1.013 | 1.005 |
| EU14 | 1.005 | 1.000 | 1.005 | 1.000 | 1.000 |
| EU27 | 1.005 | 1.023 | 0.983 | 1.013 | 1.010 |

Note: MI: Malmquist index, TEC: technical efficiency change, TC: technical change, PETC: pure technical efficiency change, SEC: scale efficiency change. Source: Own elaboration.

It is necessary to refer to the fact that a value less than 1 of the Malmquist index or any of its components means that it fell over time and just the opposite if the data are greater than equal. In other words, what has been said represents a regression or an advance of its relative position, respectively. In the same way, it should not be forgotten that the scores obtained are obtained based on the set analysed and the variables chosen, which means that a change in the group of countries subjected to evaluation or the magnitudes used will mean that the indices obtained are different.

As can be seen, productivity grew, on average, by 1.5 per 100 during the period 2000–2019 in the CEECs. This increase was due, above all, to improvements in efficiency over these years, since in terms of capitalization, transfer and innovation processes, there was a slight decrease of −0.4 percent. If the result obtained by the PECOs is compared with the EU14 group, clear differences can be seen. In this case, in this group of EU14 countries, we can see that their productivity only grew by a scale of one third (0.5 per 100) of the increase experienced by the members of the CEEC group. In addition to this, in this case, behaviour was also different in both groups. While in the CEE countries, technical change lessened the boost that the change in technical efficiency provided to aggregate productivity growth, in the EU14 group, technical change was precisely the only one responsible for the increase in the Malmquist index or, in other words, the growth of total factor productivity (TFP), since the value of the change in technical efficiency was equal to previous, exactly the same as its two components (pure technical efficiency and efficiency of scale). Looking at the last row of Table 2, which shows the values reached by the EU27 as a whole, it is easy to see that its values are in line with those obtained by the CEECs. In other words, the main protagonist of the increase in productivity (0.5 per 100) is the change in technical efficiency (2.3 per 100), while technical change (−1.7 per 100) contributed negatively to the growth of total factor productivity[11].

The analysis by country provides some additional information that should be highlighted. Of the 13 countries that make up the group that is the main research object of this work, 8 (Latvia, Bulgaria, Slovakia, Malta, Lithuania, Slovenia, Romania and Hungary) increased their TFP above average and, consequently, more than in the whole of the EU27, while only 4 countries (Czech, Poland, Cyprus and Estonia) increased their productivity at a rate below average. Only Croatia saw its TFP decrease in these years, as evidenced by the value of 0.986 obtained for the average of its Malmquist index, also being the only country with values below equal, not only in the aggregate indicator, but also in each and every one of its components.

Slovakia, Malta and Slovenia, in addition to being the countries with the highest growth in their Malmquist indices, have a similar behaviour in their components. In all three, TFP growth is explained both by the improvement in technical efficiency and by technical change, although the latter component has a much smaller contribution. As regards the increase in technical efficiency, it is necessary to highlight the much greater weight that the change in pure technical efficiency has compared to the lesser importance of scale efficiency, especially in Slovakia and Slovenia (although in Slovakia the scaling efficiency decreased by 0.3 percent). What this technological change (which also occurred in Cyprus, although its Malmquist index growth was below average) shows is that these four countries (Slovakia, Malta, Slovenia and Cyprus) managed to shift their frontier in the direction of the optimal frontier, while the rest (having a technical change indicator of less than one) moved away. Czech and Estonia, with a value equal to 1 of said technical change, remained in the same relative position.

Table 3 shows the accumulated values of the Malmquist index and its components. This information allows us to point out some issues that can be observed in the aforementioned table. In line with what was seen in the average Malmquist indices, it can be seen that, compared to growth in said indicator of less than 11% in both the EU27 and the EU14, the group of CEECs approximately tripled their growth of productivity experienced in the 21st century in relation to their western neighbours and even the whole of the EU, which,

to a large extent, is explained by the less advantageous starting situation of these countries, which allowed them to reach faster growth than their more advanced partners.

**Table 3.** Malmquist index and its components, 2000–2019. (Cumulative growth.)

| Country/Area | MI | TEC | TC | PETC | SEC |
|---|---|---|---|---|---|
| Bulgaria | 1.606 | 2.161 | 0.743 | 1.854 | 1.165 |
| Croatia | 0.765 | 0.931 | 0.822 | 0.943 | 0.986 |
| Cyprus | 1.033 | 0.992 | 1.042 | 1.000 | 0.992 |
| Czech | 1.205 | 1.212 | 0.994 | 1.156 | 1.049 |
| Estonia | 1.048 | 1.058 | 0.991 | 0.893 | 1.185 |
| Hungary | 1.377 | 1.443 | 0.954 | 1.322 | 1.092 |
| Latvia | 1.883 | 1.995 | 0.943 | 2.009 | 0.993 |
| Lithuania | 1.450 | 1.528 | 0.949 | 1.537 | 0.994 |
| Malta | 1.520 | 1.435 | 1.059 | 1.000 | 1.435 |
| Poland | 1.192 | 1.370 | 0.870 | 1.096 | 1.249 |
| Romania | 1.416 | 1.906 | 0.743 | 1.446 | 1.318 |
| Slovakia | 1.592 | 1.562 | 1.019 | 1.640 | 0.953 |
| Slovenia | 1.431 | 1.341 | 1.067 | 1.325 | 1.012 |
| CEECs | 1.315 | 1.412 | 0.932 | 1.282 | 1.101 |
| EU14 | 1.106 | 1.000 | 1.106 | 1.000 | 1.000 |
| EU27 | 1.108 | 1.529 | 0.724 | 1.273 | 1.201 |

Source: Own elaboration.

In the same way, the countries that experience higher than average TFP growth coincide with those that were in the same position when it came to the average values, although it is necessary to note some small differences in the contribution of the different components to the increase in said productivity, measured through the Malmquist index. Thus, for example, in the cumulative indices, the change in technical efficiency (TEC) as well as the average values now have values less than 1 in Croatia, as well as Cyprus; the technical change in both Czech and Estonia is below 1 and in the change in scale efficiency in Czech, Latvia and Lithuania is below 1. The rest of the values are in the same direction as those achieved as values means of the Malmquist index and its components between 2000 and 2019.

We will now analyse the aggregate behaviour of the economies of the Central and Eastern European countries belonging to the European Union in terms of total factor productivity (TFP) over these 20 years and present the results below, since one of the main objectives of this study is to determine whether the evolution of the service sector in these same countries follows the same pattern or, on the contrary, has appreciable differences depending on the economic activity in question.

The first step we took, as previous, was to study the evolution of production in the services sector and the factors that determine the level of product achieved in comparison with the manufacturing[12] sector. Table 4 shows the average annual growth rates of GVA, employment and the capital stock for the manufacturing and services sectors between 2000 and 2019, both inclusive. The first thing that can be highlighted from the data offered in said table is the strong growth in the GVA of manufacturing (7.86 percent) in the CEECs compared to that experienced by services in those same years (4.35 percent), as well as the higher growth of the manufacturing capital stock (3.74 percent) compared to that corresponding to tertiary activities (2.49). This behaviour contrasts what happened in EU14 and EU27, areas in which services recorded faster growth rates than manufacturing. This is logical when it comes to economies with a higher level of capitalization and, above all,

when referring to countries in which the degree of outsourcing and the incorporation of new technologies to the most dynamic services sectors have led to both their levels of production as well as capital accumulation growing faster than manufacturing.

**Table 4.** Evolution of production growth, labour factor and capital stock in manufacturing and services, 2000–2019. (Average annual growth rate, in %).

| Country/Area | Production | | Employment | | Capital | |
|---|---|---|---|---|---|---|
| | **Man** | **Ser** | **Man** | **Ser** | **Man** | **Ser** |
| Bulgaria | 5.98 | 5.52 | −1.25 | 1.42 | 4.16 | 2.09 |
| Croatia | 8.45 | 3.16 | −0.28 | 0.75 | 4.68 | 1.80 |
| Cyprus | 6.76 | 4.59 | −0.27 | 0.68 | 9.70 | 2.74 |
| Czech | 0.93 | 3.03 | −1.26 | 1.31 | −0.07 | 3.76 |
| Estonia | 0.91 | 3.91 | −1.05 | 2.65 | −0.09 | 2.71 |
| Hungary | 3.19 | 4.92 | −0.25 | −0.32 | 3.05 | 0.58 |
| Latvia | 9.18 | 5.46 | −1.45 | 1.07 | 4.27 | 4.56 |
| Lithuania | 3.35 | 3.47 | −0.42 | 1.06 | 5.51 | 1.68 |
| Malta | 0.29 | 8.98 | −0.67 | 8.73 | 1.00 | 4.27 |
| Poland | 12.52 | 4.86 | −0.24 | 1.09 | 4.84 | 3.78 |
| Romania | 5.94 | 5.58 | −1.41 | 1.70 | 1.54 | 1.98 |
| Slovakia | 4.45 | 3.18 | −1.19 | 1.86 | 0.29 | 0.38 |
| Slovenia | 18.85 | 3.34 | −0.73 | 1.10 | 4.54 | 1.68 |
| CEECs | 7.86 | 4.35 | −0.81 | 1.17 | 3.74 | 2.49 |
| EU14 | 1.29 | 1.66 | −0.65 | 1.06 | 0.66 | 1.11 |
| EU27 | 1.74 | 1.85 | −0.70 | 1.09 | 0.93 | 1.21 |

Note: Man: manufacturing; Ser: services. Source: Eurostat and EU KLEMS. Own elaboration.

An additional question is related to the evolution of employment. In this sense, it can be noted that in all the countries under study, the volume of employment decreased in manufacturing, while it grew in services (except in Hungary), which may be an indicator of the increase in employment productivity of the labour factor in manufacturing and a decline in services, which is fully compatible with the existing literature on the productivity of the productive sectors and, especially, with the stylised facts on the evolution of the productivity of services, usually characterised (although there are branches of the tertiary sector to which this cannot be applied) by slow growth and even decrease in their productive efficiency.

The analysis of the evolution of the TFP can be performed from the data shown in Table 5, in which the values of the Malmquist index, distinguishing between manufacturing industrial activities and services, as well as the different components into which the aforementioned index can be broken down can be seen, in accordance with the proposal made by Färe et al. (1994).

The first observation that can be highlighted from this table is that, in general, the increase in productivity in the manufacturing sector exceeds that of the tertiary activities of the economy. The group of countries studied experienced an average growth of their TFP of 1.6 compared to 1.3 per 100 registered by services. This means that the productivity of industry increased in all these years at a faster rate than the economy as a whole, while services were below said average values. Despite this, some countries had productivity growth in the services sector that was higher than that of manufacturing, such as Hungary, Bulgaria, Latvia, Malta and Estonia, and in the latter two there was even a decline in the productivity of the industrial sector in this century. In any case, except for Croatia, where the productivity of services fell, in all the CEECs, there was an increase in tertiary productivity higher than that registered in the countries that were already part of the EU

when they joined it (UE14); this increase was even higher than that of the total group, that is to say, the UE27.

**Table 5.** Malmquist index and its components in manufacturing and services, 2000–2019. (Average annual growth).

| Country/Area | MI | | TEC | | TC | | PETC | | SEC | |
|---|---|---|---|---|---|---|---|---|---|---|
| | Man | Ser | Man | Ser | Man | Ser | Man | Ser | Man | Ser |
| Bulgaria | 1.009 | 1.020 | 1.009 | 1.012 | 1.001 | 1.008 | 1.009 | 1.015 | 0.999 | 0.998 |
| Croatia | 1.009 | 0.998 | 1.009 | 0.988 | 1.001 | 1.010 | 1.006 | 0.991 | 1.003 | 0.997 |
| Cyprus | 1.009 | 1.008 | 1.010 | 0.999 | 0.999 | 1.008 | 1.016 | 1.000 | 0.995 | 0.999 |
| Czech | 1.019 | 1.011 | 1.019 | 1.000 | 1.001 | 1.010 | 1.015 | 1.003 | 1.003 | 0.997 |
| Estonia | 0.989 | 1.014 | 0.990 | 1.003 | 0.998 | 1.011 | 0.991 | 1.000 | 1.000 | 1.003 |
| Hungary | 1.010 | 1.015 | 1.000 | 1.003 | 1.010 | 1.012 | 1.000 | 1.004 | 1.000 | 0.999 |
| Latvia | 1.002 | 1.032 | 1.000 | 1.020 | 1.002 | 1.011 | 1.000 | 1.021 | 1.000 | 0.999 |
| Lithuania | 1.023 | 1.007 | 1.021 | 0.997 | 1.002 | 1.010 | 1.023 | 1.000 | 0.998 | 0.998 |
| Malta | 0.994 | 1.013 | 0.993 | 1.008 | 1.000 | 1.005 | 1.000 | 1.000 | 0.993 | 1.008 |
| Poland | 1.032 | 1.007 | 1.037 | 0.998 | 0.995 | 1.009 | 1.011 | 1.000 | 1.025 | 0.998 |
| Romania | 1.027 | 1.021 | 1.026 | 1.010 | 1.001 | 1.010 | 1.024 | 1.013 | 1.002 | 0.998 |
| Slovakia | 1.051 | 1.012 | 1.052 | 1.002 | 0.999 | 1.010 | 1.051 | 1.004 | 1.001 | 0.997 |
| Slovenia | 1.030 | 1.020 | 1.029 | 1.009 | 1.001 | 1.010 | 1.026 | 1.009 | 1.003 | 1.001 |
| CEECs | 1.016 | 1.013 | 1.015 | 1.004 | 1.001 | 1.010 | 1.013 | 1.005 | 1.002 | 0.999 |
| EU14 | 1.006 | 1.005 | 1.004 | 1.000 | 1.001 | 1.005 | 1.000 | 1.000 | 1.004 | 1.000 |
| EU27 | 1.011 | 1.007 | 1.063 | 1.012 | 0.951 | 0.995 | 1.038 | 1.007 | 1.023 | 1.006 |

Source: Own elaboration.

Disaggregating TFP growth into its two main components, it can be seen how that corresponding to technical change is the main cause of the boost experienced by productivity in the 21st century, both in the CEECs and the EU14 and EU27, when we refer to the services sector, although this was not the case in Malta, Bulgaria and Latvia, countries in which the change in technical efficiency was the main driver of the increase (or decrease in the particular case of Malta) in total productivity of the factors.

This behaviour is clearly different from what happened in the industrial sector (as well as in the economy as a whole), where, in most of the areas included in Table 5, it can be seen that the increase in the technical efficiency with which production is obtained, the most important component for the change in TFP in the manufacturing sector (with the exceptions of Estonia and Malta, where the TEC fell, and Hungary and Latvia, where it remained unchanged, that is, 1, which means that in these four countries, technical change caused the increase in TFP in the industrial sector).

The change in total factor productivity of services in the CEECs could be explained, for the most part, by technical change; however, it is also of interest to find out what factors are behind the increase (or decrease) in technical efficiency, even if it is of less magnitude and, therefore, its contribution to the variation of the Malmquist index is less. Changes in technical efficiency depend on the economic sector in question, as has been said, since they are more important in industry than in services. However, regardless of this, it would be necessary to see if pure technical efficiency or scale efficiency contribute to a greater or lesser extent to technical efficiency. In this sense, it can be pointed out that the variation in pure technical efficiency takes on a greater role than scale efficiency in determining the growth of technical efficiency, regardless of whether we refer to manufacturing or services (with some exceptions, obviously).

What this shows is that the change in scale had a reduced influence on the growth of technical efficiency. However, it should be noted that it was the only factor responsible for the growth of technical efficiency in manufacturing in the EU14.

The information contained in Table 6, as was calculated for the economy as a whole in the corresponding Table 3, offers some values that need to be briefly elaborated. The first thing is that, as in the case of the average values, in the accumulated values of the TFPs of manufacturing are also higher than those of the economy as a whole and those of services are below said average value. This highlights the gap between manufacturing and services in the CEECs, which rose to 5.2 points (34.1 versus 28.9 per 100, respectively) and, in turn, the notable differences that they maintain with the group of countries that they were already part of the EU when they joined (EU14). In this last group, the variation in TFP is only due to technical change when talking about services. Just as this variation happened when analysing the accumulated values of the economy as a whole, it happens when performing analysis on a sectoral basis, that is, the general behaviour is of the same nature.

**Table 6.** Malmquist index and its components in manufacturing and services, 2000–2019. (cumulative growth).

| Country/Area | MI | | TEC | | TC | | PETC | | SEC | |
|---|---|---|---|---|---|---|---|---|---|---|
| | **Man** | **Ser** | **Man** | **Ser** | **Man** | **Ser** | **Man** | **Ser** | **Man** | **Ser** |
| Bulgaria | 1.194 | 1.467 | 1.176 | 1.262 | 1.015 | 1.162 | 1.194 | 1.317 | 0.985 | 0.958 |
| Croatia | 1.194 | 0.954 | 1.179 | 0.791 | 1.012 | 1.206 | 1.117 | 0.841 | 1.056 | 0.941 |
| Cyprus | 1.193 | 1.154 | 1.216 | 0.985 | 0.981 | 1.171 | 1.348 | 1.000 | 0.902 | 0.985 |
| Czech | 1.434 | 1.222 | 1.418 | 1.003 | 1.011 | 1.218 | 1.328 | 1.053 | 1.068 | 0.953 |
| Estonia | 0.806 | 1.299 | 0.832 | 1.055 | 0.970 | 1.231 | 0.838 | 1.000 | 0.992 | 1.055 |
| Hungary | 1.199 | 1.323 | 1.000 | 1.063 | 1.199 | 1.244 | 1.000 | 1.085 | 1.000 | 0.980 |
| Latvia | 1.045 | 1.804 | 1.000 | 1.463 | 1.045 | 1.234 | 1.000 | 1.495 | 1.000 | 0.978 |
| Lithuania | 1.536 | 1.144 | 1.490 | 0.953 | 1.031 | 1.200 | 1.552 | 0.992 | 0.960 | 0.960 |
| Malta | 0.887 | 1.276 | 0.883 | 1.166 | 1.004 | 1.095 | 1.000 | 1.000 | 0.883 | 1.166 |
| Poland | 1.823 | 1.140 | 1.995 | 0.963 | 0.914 | 1.184 | 1.240 | 1.000 | 1.609 | 0.963 |
| Romania | 1.653 | 1.471 | 1.620 | 1.218 | 1.020 | 1.208 | 1.556 | 1.275 | 1.041 | 0.955 |
| Slovakia | 2.556 | 1.252 | 2.614 | 1.031 | 0.978 | 1.214 | 2.575 | 1.082 | 1.015 | 0.953 |
| Slovenia | 1.749 | 1.444 | 1.725 | 1.191 | 1.014 | 1.213 | 1.634 | 1.178 | 1.056 | 1.011 |
| CEECs | 1.341 | 1.289 | 1.323 | 1.076 | 1.013 | 1.198 | 1.282 | 1.090 | 1.032 | 0.987 |
| EU14 | 1.114 | 1.090 | 1.089 | 1.000 | 1.023 | 1.090 | 1.000 | 1.000 | 1.089 | 1.000 |
| EU27 | 1.221 | 1.141 | 3.173 | 1.266 | 0.385 | 0.902 | 2.050 | 1.132 | 1.548 | 1.118 |

Source: Own elaboration.

## 5. Conclusions

The role that productivity plays in long-term economic growth is sufficiently evidenced. The purpose of this work was to measure the changes in productivity in the countries that joined the European Union later, which are also those countries that have a somewhat lower level of development than the states that were part of the European Union at the end of the last century.

The fact that the services sector is the one that contributes the most to the production of the advanced countries and the repercussion that the variation in its productivity has on growth is what has motivated the realization of this work, because if the productivity of this sector (or its variation) is slow (or null), given the role it has in the economy (more than 60 percent of the GDP of each of the EU countries), this will determine if the growth of said country is greater or lesser.

The objectives set out in this work were to study how the productivity of services in the CEECs evolved so far this century and ascertain the factors that explain its evolution, that is, if the advance of techniques was decisive or, instead, the improvement in the efficiency with which services are produced was most responsible for this variation; in addition, we sought to determine if the change in said efficiency (assuming that it occurred) was caused by a change in scale efficiency or instead by a variation in pure technical efficiency.

Before analysing what happened in the services sector, an evaluation of the behaviour of productivity for the economy as a whole (with all sectors) was carried out. From this, a couple of conclusions were obtained that highlight, on the one hand, that the productivity of the CEECs grew three times that of the EU14 and, on the other, that in the CEECs, technical efficiency was what caused TFP growth, while in the EU14 technical change was the only one responsible for the increase in that productivity.

The analysis of the services sector and its comparison with the manufacturing sector, since this is usually the economic sector with the greatest increase in productivity, allowed us to extract a series of findings: (a) productivity growth was higher, in general, in industry than in services; (b) the increase in productivity in services in the CEECs exceeded that obtained by the EU14 and EU27; (c) the main cause of productivity growth in services in the CEECs was technical change (the same as in the EU14 and EU27), unlike what happened in the economy as a whole and the manufacturing sector, in which it was the improvement in technical efficiency that was responsible for said growth; (d) the variation in the technical efficiency of the CEECs (although less relevant than the technical change in the case of services) was due, fundamentally, to pure technical efficiency, as was also produced in the case of manufacturing; and (e) in the EU14 the change in efficiency of scale was the only factor that explains the improvement in technical efficiency in the industry.

In short, the factors that determine growth in the productivity of services are different from those of manufacturing and, likewise, there are differences in the factors that determine said variations between the countries with a somewhat lower level of development (CEECs) and those that are more advanced (UE14).

The results obtained, in short, show that in the CEECs, the service sector can continue to gain in productivity, as it started from a lower level of development, which is explained by the lesser importance of tertiary activities in relation to GDP, compared to the EU14. This means that they still have room for productivity gains until they reach a level equivalent to that of the EU14, and thereafter, these productivity increases are likely to be moderate as in the more advanced countries, since the difference lies in the greater possibility of introducing technical changes, the only factor responsible for the growth in productivity of services in both groups. In other words, at the policy level, services growth and efficiency improvements should be further encouraged in order to contribute to aggregate productivity improvements in the CEECs.

Before concluding this paper, it is necessary to refer to some of its limitations. On the one hand, the period chosen could be different and the results would change to some extent. On the other hand, the grouping of countries was made according to the time of their membership within the EU, so that if the selection had been made in another way, for example, by homogeneous groups according to the weight of the service sector, eliminating those in which this activity was less representative, the results could also be different. Thirdly, the methodology used, although appropriate as has been shown by its use in other studies, may offer different conclusions to those obtained by other techniques. In any case, changing these aspects would have entailed performing a different work from this one, also with its own limitations. By this, we simply want to emphasise that the results obtained respond to the variables, the spatial and temporal spheres studied, in such a way that changes in any of them would imply a modification of the values obtained. For example, if instead of analysing the service sector as an aggregate, we had broken it down into different branches of activity, it could be seen that there are branches whose Malmquist index exceeds that of manufacturing and others with values below it (Maroto 2009), but that is beyond the scope of this paper.

Finally, some aspects that could be considered for future work would be, for example, on the one hand, those that take into account the effect that COVID-19 may have had on productivity (obviously, when the availability of data allows it) and, on the other, the breakdown of the services sector into its branches of activity (according to the A10 classification of NACE Rev.2, for example), to the extent that the data allow it, since it is logical to think that the behaviour of services is not the same in all the activities it integrates and, therefore, the determinants of productivity growth and their effects will be different.

**Author Contributions:** Conceptualization, A.A.-O., F.A.-O. and P.J.C.-G.; methodology, F.A.-O.; validation, F.A.-O. and P.J.C.-G.; formal analysis, A.A.-O. and F.A.-O.; investigation, A.A.-O. and P.J.C.-G.; resources, P.J.C.-G.; writing—original draft preparation, A.A.-O., F.A.-O. and P.J.C.-G.; writing—review and editing, P.J.C.-G.; visualization, A.A.-O., F.A.-O. and P.J.C.-G.; supervision, A.A.-O., F.A.-O. and P.J.C.-G.; project administration, P.J.C.-G.; funding acquisition, A.A.-O. All authors have read and agreed to the published version of the manuscript agreed to the published version of the manuscript.

**Funding:** This research received no external funding.

**Informed Consent Statement:** Not applicable.

**Data Availability Statement:** Not applicable.

**Conflicts of Interest:** The authors declare no conflict of interest.

## Notes

1.   A summary of the main theoretical contributions on the relationship between the services sector and the evolution of aggregate productivity can be found in Cuadrado and Maroto (2012); however, a more detailed review of these contributions can be found in Maroto (2009).

2.   Some empirical contributions in this regard in Europe can be found in O'Mahony and van Ark (2003) and van Ark and Piatkowski (2004). For their part, some works such as those by Stiroh (2002) and Triplett and Bosworth (2006) focus on the American case.

3.   The Malmquist productivity index, which is a multi-factor productivity index, comprises three indices, namely technology change index, technical efficiency change index and scale efficiency change index, and is a robust and appropriate measure of sustainable, multifactor productivity (Ambarkhane et al. 2019).

4.   As Walheer (2022) points out, some authors such as O'Donnell (2012) and Peyrache (2014) have questioned the suitability of this method. However, that is a debate that is outside the scope of this work.

5.   In Section 3, the decomposition of the MI that was used in this work is exposed, as well as its meaning and interpretation.

6.   The results and conclusions obtained in the case of production per worker are essentially the same, as is also indicated in the work of Cuadrado and Maroto (2007).

7.   Farrell (1957) divided efficiency into technical and price efficiency. The former reflects the ability of a firm to obtain maximal output from a given set of inputs, while the latter shows the ability of a firm to use the inputs in optimal proportions, given their respective prices and the current production technology. The sum of both types of efficiency is called global economic efficiency.

8.   The reason for choosing an output-oriented DEA model is based on the fact that the efficiency of the service sector would reside in obtaining the maximization of production in the provision of the different services without modifying the amount of input used, which would be determined by the needs of each activity. In this context, it would not make sense to evaluate the services from the point of view of minimizing the input with the same level of output.

9.   Some representative works that have used different techniques and/or have proposed some decomposition of the Malmquist index to allow the analysis of its sources of growth are those of Färe et al. (1992), Ray and Desli (1997), Coelli et al. (1998), Balk (2001), Rossi (2001), Fuentes et al. (2001), Orea (2002), Lovell (2003), Grosskopf (2003) and Pantzios et al. (2011), among others.

10.   The data of the rows of Table 1 and the following ones correspond to the different countries of Central and Eastern Europe, although at the end of each of the data tables presented, it was decided to include, in addition to the average of this block of countries (CEECs), the one corresponding to those who were in the EU before the incorporation of these 13, excluding the United Kingdom, since it already left the EU; the average of all the countries belonging to the current European Union (EU27) is also offered.

11.   When referring to a more specific activity in the service sector, such as hotel establishments, the meaning of each of the components of the Malmquist index would be as follows Frančeškin and Bojnec (2022): Technical change (TC) involves the application of new technologies to production that increase productivity; these are the increases that occur in output due to innovations introduced in the process. Pure technical efficiency change (PETC) reveals investment in organisational factors associated with hotel management, such as marketing initiatives, quality improvements and a better balance between inputs and outputs. The

change in scale efficiency (SEC) reveals changes in the size of hotel companies. Finally, the change in technical efficiency (TEC) represents the improvement in efficiency due to the two previous factors.

[12]  In accordance with the A10 classification, NACE Rev.2, section C, the manufacturing industry will be considered as industry and sections ranging from G to U, both inclusive, as services.

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
