# Peer review of "Productivity of Services in the Countries of Central and Eastern Europe: Analysis Using Malmquist Indices"

_economies, doi:10.3390/economies11030091_

Round 1

Reviewer 1 Report

The study is basically consistent, proposing the appropriate research method and presenting interesting results. However, there is a place for improvement:

In the abstract, the geographical scope of the research (Central and Eastern Europe) should be added or discussed.

A table or a graph for country ranking can be added to the 4. Results and discussion based on the indicators.

Limitations and policy implications are missing in the Conclusion. Please discuss it.

Reviewer 2 Report

Dear authors, thank you for the opportunity to read your research!

The article is devoted to the analysis of service productivity in the countries of Central and Eastern Europe using non-parametric methods. The relevance of this study is primarily due to the use of the DEA methodology and the Malmquist index, which are of great research interest when applied to various objects of economic activity.

The article is well structured, the purpose of the study is clearly defined and methodologically justified. All sources used are relevant and applicable to the manuscript. All tables in the manuscript are informative, help to understand the reasoning and conclusions of the authors.

Highly appreciating the presented manuscript, I have a few comments to the authors:

1. The authors consider in detail the DEA methodology and the Malmquist index, but do not indicate in which software the resulting DEA indicators and the Malmquist index were calculated.

2. The authors present the average annual Malmquist Index and its components in the service sector: technical efficiency change, technical change, pure technical efficiency change, scale efficiency change. For the understanding of readers who are not familiar with the practical application of the DEA methodology, it is necessary to interpret the results obtained not only on the example of individual countries (this is in the manuscript), but also on the example of specific sub-sectors / activities in the service sector. It is necessary to give an example of what technical efficiency, technical change, pure technical efficiency change, scale efficiency change, for example, in the field of tourism or in the field of education (or any other example).

3. Based on a comparison of the calculated Malmquist Indices and its components in the manufacturing industry and the service sector, the authors conclude that productivity in the service sector is growing at a much slower rate than in the industrial sector. But can the growth rate of technical efficiency in the service sector outstrip the growth rate of technical efficiency in industry, given that services have such characteristics as intangibility, inseparability, impermanence, non-persistence, which (in some activities) cannot be automated and / or cannot be separated from the person providing the service?

4. The authors should expand the conclusions: indicate what the results mean for the service sector in the countries of Central and Eastern Europe and / or for specific types of service sector activities; how the results can be used.

Round 2

Reviewer 2 Report

The authors have done work to enrich the text of the manuscript. I have no comments to the authors of the manuscript.